# Mothers in Need of Lactation Support May Benefit from Early Postnatal Galactagogue Administration: Experience from a Single Center

**DOI:** 10.3390/nu14010140

**Published:** 2021-12-29

**Authors:** Eleni Karapati, Alma Sulaj, Adamantia Krepi, Abraham Pouliakis, Nicoletta Iacovidou, Stella Paliatsiou, Rozeta Sokou, Paraskevi Volaki, Theodora Boutsikou, Zoi Iliodromiti

**Affiliations:** 1Neonatal Department, Medical School, National and Kapodistrian University of Athens, Aretaieio Hospital, 11526 Athens, Greece; helenak5@hotmail.com (E.K.); alma_sulaj@hotmail.com (A.S.); mandokr1@gmail.com (A.K.); niciac58@gmail.com (N.I.); stpaliatsiou@yahoo.gr (S.P.); sokourozeta@gmail.com (R.S.); v.volaki@hotmail.com (P.V.); theobtsk@gmail.com (T.B.); 2Second Department of Pathology, Medical School, National and Kapodistrian University of Athens, University General Hospital Attikon, 12462 Athens, Greece; apou1967@gmail.com

**Keywords:** breastfeeding, galactagogues, lactation, Silitidil, breastfeeding rates, neonate

## Abstract

Background: Galactagogues are substances that promote lactation, although data on their effects on humans remain limited. We investigated the efficacy of Silitidil to increase milk supply and duration of breastfeeding of a specific subgroup of mothers in need of lactation support. Methods: 161 mothers from November 2018 until January 2021 were the study subjects in this retrospective study; during their hospitalization, due to neonatal or maternal factors that inhibited lactation, they were prescribed galactagogues. Mothers were surveyed by telephone interview via a 13-item questionnaire. Results: 73.91%, were primigravidas, 78.26% gave birth by cesarean section (CS) and 72.05% continued to take galactagogues after hospital discharge. Of the neonates, 24.22% were preterm ≤37 weeks of gestation, and 55.9% had birth weight (BW) between 2500 and 3500 g. With respect to breastfeeding rates, 100% were breastfed during their first week, 98.8% breastfed during the first month, 87% during the first 4 months, dropping to 56.5% at 6 months, 41% at 1 year and 19.3% over 1 year of age. Conclusions: This study demonstrates that administration of a galactagogue containing Silitidil (Piulatte-Humana) improves breastfeeding rates at from 1 until 12 months of life in mothers with low milk supply during their hospital stay. Further studies are needed to generate evidence-based strategies to improve breastfeeding outcomes.

## 1. Introduction

The World Health Organization (WHO) recommends that all infants be exclusively breastfed until 6 months of age and that breastfeeding continue as an important element of their diet until 2 years of age [1]. Human milk (HM) is rich not only in proteins and lipids but also in carbohydrates, especially human milk oligosaccharides (HMOs), which help the growth of beneficial bacteria and regulate mucosal and systemic immune functions [1,2]. Furthermore, HM contains bioactive components such as lactoferrin, leukocytes, secretory immunoglobulin (IgA), epidermal growth factor, hormones (insulin, leptin, ghrelin) and microRNAs [1]. These bioactive components, through the formation of beneficial microbiome and epigenetic mechanisms, modulate the immunologic response and metabolic profile of humans throughout life [3,4]. Breastfeeding provides protection against several infectious, atopic and cardiovascular diseases, as well as necrotizing enterocolitis, leukemia, celiac disease, inflammatory bowel disease and developmental and behavioral disorders [2].

In Greece, high rates of breast-feeding initiation (94%) and the rates of any breast-feeding (about 80% during the 1st month, reaching 45% at the end of the 6th month postpartum) have recently been reported [5]. However, despite the benefits of breastfeeding, the rates of exclusive breastfeeding in Greece still remain low. The reported rates for exclusive breastfeeding during the 1st month ranged from about 40% to almost <1% by 6 months of age [5]. The most common reason for discontinuation of breastfeeding is lactation insufficiency (commonly known as «low breast milk supply»). In such a situation, in addition to non-pharmacological interventions, galactagogues may also be used.

Galactagogues are substances that promote lactation, and they are categorized as herbals or pharmaceuticals. The most commonly used pharmaceutical galactagogues are domperidone and metoclopramide [6,7]. They increase prolactin secretion by antagonizing dopamine [7,8,9]. Because of their adverse effects (arrhythmias, cardiac arrest, anxiety, depression and sedation), herbal supplements are usually preferred. The most commonly used herbs are fenugreek, goat’s rue, milk thistle (Silybum marianum), oats, dandelion, millet, seaweed, anise, basil, blessed thistle, fennel seeds, and marshmallow [6]. Herbal galactagogues lack standard dosing preparation, and could potentially be allergenic or interact with other drugs [10].

Silymarin, a component of milk thistle (Silybum marianum), is a natural galactagogue and has been reported to be used since the 4th century BC. Theophrastus was the first to refer to it as “Pternix”, and later Dioskurides referred to it in his “Materia medica” [11]. Silymarin includes four flavonolignans: silybin (65%), silychristin (20%), silydianin (10%) and isosilybin (5%). Due to its low bioavailability when given orally, silymarin from thistle is combined with phospholipid biotechnologically, in order to increase its absorption. Data on the effects of silymarin on lactation are scarce. Studies in animal models, such as cows, pigs and rats showed that silymarin increased the levels of prolactin but had no effect either on mammary gland development, or on plasma progesterone or estradiol. Thus, it remains inconclusive whether there was an effect on milk production. Data regarding humans are still scarce [12,13,14].

The aim of this study was to assess whether the administration of Silitidil (Silymarin from thistle with incorporated phospholipids) was able to initiate/establish, maintain and increase milk production and influence the duration of breastfeeding of a specific population of mothers that had a higher risk of not achieving exclusive breastfeeding according to recent studies [15,16], such as twin pregnancies, prematurity or severe illness of the mother, or mothers needing additional lactation support, for example, when excessive weight loss of the infant or low milk supply during hospital stay via pumping was observed, etc. Galactagogue administration could improve breastfeeding rates in this specific group of mothers and possibly enable them to reach the breastfeeding rates of the general population.

## 2. Materials and Methods

This retrospective study included 161 mothers who gave birth at Maggineio Maternity Clinic, Aretaieio Hospital, Athens, between November 2018 and January 2021. During their hospital stay, all mothers that had a higher risk of not achieving exclusive breastfeeding according to recent studies [15,16], mothers needing additional lactation support, or mothers with low milk supply observed while pumping were prescribed Silitidil (a daily single dose of 5 g of Piùlatte by Humana). This population included, more specifically, mothers of twins or premature newborns, or those whose neonates had (a) weight loss ≥10% of BW, (b) needed phototherapy, (c) required transport to a tertiary NICU, and finally mothers unable to breastfeed due to any other reason. The 5 g of galactagogue was offered for free to the mothers that were included in the study during their hospital stay. After that period of time, we prescribed the galactagogues and they were purchased by the mothers. The prescribed galactagogues were taken once daily for 14 days. Mothers were surveyed by telephone interview at 10 days, 1, 4 and 6 months via a 13-item questionnaire that included information on the infant’s weight, the use of breast pump at home and the type of feeding. As far as the weight of the infant was concerned, the data were extracted by the mothers from the personal health record of the infant at those specific age milestones, when according to national guidelines by the institute of child health, infants are vaccinated and clinically evaluated by a certified health care professional. All mothers provided informed consent. The interviewers evaluated any possible adverse effects of Silitidil as well. Infant demographics included gestational age (GA), BW, gender, and delivery mode. Clinical data regarding parity, nationality, skin-to-skin contact and the reason for Silitidil administration were recorded.

### Statistical Methods

Statistical analysis was performed using the SAS for Windows 9.4 software platform (SAS Institute Inc., Cary, NC, USA). Test for normality was performed via the Kolmogorov–Smirnov test; birth weight and maternal age were found to be derived from a normal distribution, thus the descriptive values were expressed as mean ± SD (standard deviation). Categorical data were presented as frequency of occurrence and the relevant percentages. Weight gain in the 1st, 4th and 6th months was with reference to the birth weight.

## 3. Results

Detailed demographic data regarding the nationality of the mothers are summarized in Table 1. Of the study group, 68.94% were native Greek, while 15.53% and 5.59% were of Albanian and Asian origin, respectively; 73.91% were primigravidas and 78.26% gave birth by CS (Table 1). Additionally, 72.05% of the mothers continued to take galactagogues after hospital discharge. 

As for the neonates, 40.99% were female and 59.01% were male (Table 2), 55.9% had BW between 2500 and 3500 g, 18.63% were of low birth weight (LBW) ≤ 2500 g, and the remaining 25.47% had BW ≥ 3500 g. Further, 24.22% of the neonates were preterm ≤37 weeks of gestation, and the rest were full term (Table 2). 

The reasons for galactagogue administration to the mothers were as follows: (a) for 58.39% of the study population, the neonate’s weight loss exceeded 10% of BW, (b) 13.66% were twin pregnancies, (c) 8.7% were preterm births, (d) 8.07% were receiving phototherapy (e) 6.21% were due to NICU admission, and finally (f) 8% were due to maternal morbidity of various etiologies (Table 3). Skin-to-skin contact had taken place only in 20.50% of births. The majority of the mothers (93.79%) used a breast pump during their hospital stay. More specifically, 50.33% followed our guidance and pumped every 3 h, 16.56% pumped 3 to 4 times per day and 29.8% pumped only once or twice per day. As per our advice, 61.49% of the mothers continued pumping at home (Table 3).

As for the first feeding of the infant, 65.2% were breastfed, 21.74% had formula and 13.04% received intravenous fluids. The reasons for formula administration were weight loss in 44.27% of cases, maternal desire in 16.03%, against our advice to the contrary, NICU admission in 11.45%, twin pregnancy in 9.16%, maternal disease of any etiology in 9.16%, and phototherapy in 9.92%. With respect to breastfeeding rates, 100% of infants were breastfed during their first week, 98.8% were breastfed during the first month, 87% during the first 4 months, 56.5% at 6 months, 41% at 1 year and 19.3% over 1 year of age (Table 4). 

## 4. Discussion

Many factors are associated with a mother’s decision to initiate and continue breastfeeding. Caesarian section is associated with decreased breastfeeding initiation and duration. This trend could be explained not only by the low rates of skin-to-skin contact after birth, but also because of delayed onset of galactogenesis, due to decreased oxytocin secretion or increased cortisol secretion because of stress imposed on the infant or because of the maternal morbidity that led to the CS [15,16]. In this survey, we recorded high rate of CS, close to 78% (due to maternal or fetal conditions), higher when compared to the national rate of 48.8% [17]. Our cohort comprised mothers who had difficulties regarding lactation and had insufficient milk production, as noted while pumping during their hospital stay. Primiparity is negatively correlated with the continuation of breastfeeding, mainly because of lack of experience and delayed galactogenesis, in comparison to multiparous women [15,18]. Our results are in line with that finding, with 74% of the mothers in need of lactation support and galactagogue administration because they were primiparous. Despite that limiting factor, we achieved high rates of breastfeeding. Neonatal disorders such as prematurity or NICU admission, or maternal morbidities such as ICU admission, lower the chances of breastfeeding [16]. Maastrup R. et al., in a study of 1488 preterm neonates, found that only 82% of their mothers breastfed at 1 month of age, and this dropped to 56% at 4 months [19]. In our study the percentage of mothers who required galactagogue administration due to prematurity of the offspring was 24%. Regarding the duration of breastfeeding in our cohort, 98.8% breastfed at 1 month of age and 87% breastfed at 4 months of age. These rates are higher than Greece’s rates in the general population, as mentioned above, and 45% of the mothers continued breastfeeding at 6 months, although the neonates in our study had lower chances of initiating and continuing breastfeeding due to their conditions. One of the most important factors associated with high breastfeeding rates is education on breastfeeding, which is achieved with prenatal breastfeeding courses and assistance from health care providers in the perinatal period [20]. As a “baby friendly hospital”, we support breastfeeding by ensuring that every first-line intervention such as a good latch-on is applied, but also by implementing additional interventions such as galactagogue administration and breast-pumping when required. We also practice rooming-in and skin to skin contact with the mother right after birth and ensure breastfeeding during the first hour of life (the golden hour). Skin-to-skin contact, especially in the first hour after birth, is linked to high rates of breastfeeding initiation and continuation [16,21,22]. It not only enhances the mother-child bond, but also promotes the early stimulation of the breast, and as such the galactogenesis and suckling reflexes of the infant. According to our findings, almost 80% of the mothers who were prescribed galactagogues had not had skin-to skin contact immediately after birth. This is a very high rate compared to our hospital’s statistics, since we are a baby-friendly hospital with an 87.2% skin-to-skin contact rate for total births (data from our statistics). This can be explained by the fact that either these neonates were of high risk or the mothers had complications during labor and were unable to provide skin-to-skin contact. Despite this limiting factor, breastfeeding rates reached 100% during the first week of life, when galactagogues were administered, a higher percentage than the 94% that is noted nationally in Greece for the same time period [5]. Due to neonatal morbidity in our cohort, the percentage of formula administration as the first feeding of the baby was high (21%) and definitely deviated from the data for our hospital, where breastfeeding is chosen as the first feeding in 94% of the neonates. One of our questions during our telephone interview was about the weight gain of the infant at the 1^st^, 4^th^ and 6^th^ month of age. Our outcomes were classified by gender and compared with global growth charts from WHO. The percentage of weight gain at these specific time-points followed the growth curve of WHO [23] (Table 5).

Regarding the possible mechanisms of action of galactagogues on lactation, to the best of our knowledge these have not yet been clarified. Experiments in mice showed an increase in prolactin production and secretion, which could justify the effect they have in milk production. Flavonolignans decrease estrogen levels in experimental tumor mice models. In this way, Silitidil promotes lactation [13,14,24].

Our study has certain limitations. First, we cannot exclude the psychological effects the galactagogues had on the mothers (placebo effect) [25]. Mothers mentioned that Silitidil made them feel “self-worthy”. They seized every opportunity to breastfeed and provide the best nutrition for their child. On the other hand, the role that the breast pump plays in lactation is widely known and cannot be excluded as a positive effect in establishing lactation and promoting increased breastfeeding duration. Finally, our cohort was heterogenic and included twin pregnancies, neonates from high-risk pregnancies and premature births. The common characteristic they shared was the lower chance of initiating and maintaining breastfeeding compared to the general population.

## 5. Conclusions

Despite its limitations, our study demonstrated that administration of a galactagogue containing Silitidil (Piulatte by Humana) improves breastfeeding rates between 1 and 12 months of life in mothers with low milk production during their hospital stay. Although there were difficulties in initiating breastfeeding, our sample achieved higher breastfeeding rates and breastfeeding duration (Table 4) compared to national rates [5]. Moreover, the percentage of weight gain at these specific time points of the neonates we studied followed the growth curve of WHO [23] in spite of the higher risk of a lower growth curve (Table 5). Our findings underline the need for further research in larger patient groups, particularly randomized controlled trials, to evaluate the efficacy and safety of galactagogues and support evidence-based strategies to improve breastfeeding outcomes.

## Figures and Tables

**Table 1 nutrients-14-00140-t001:** Baseline characteristics of the mothers.

Characteristic	Measure
Maternal age in years (mean ± SD)	34.6 ± 5.4
Nationality	
Greek	111(68.94%)
Albanian	25 (15.53%)
Asian	9 (5.59%)
From western Europe	5 (3.11%)
Romanian	5 (3.11%)
Bulgarian	2 (1.24%)
Other	4 (2.48%)
Parity	
1st	119 (73.91%)
2nd	31 (19.25%)
3rd	10 (6.21%)
4th	1 (0.62%)
Delivery mode	
Forceps use	10 (6.21%)
CS	126 (78.26%)
Vaginal delivery	25 (15.53%)

**Table 2 nutrients-14-00140-t002:** Baseline characteristics of the neonates.

Characteristic	Measure
Gender female (N,%)	66 (40.99%)
BW in grams (N,%)	
≤2500	30 (18.63%)
2500–3500	90 (55.9%)
≥3500	41 (25.47%)
GA in weeks + days (N,%)	
<37	39 (24.22%)
≥37	122 (75.78%)

**Table 3 nutrients-14-00140-t003:** Feeding characteristics in relation to the mother.

Characteristic	Measure
Reasons for galactagogue administration	
Weight loss > 10%	94 (58.39%)
Twin pregnancy	22 (13.66%)
Prematurity	14 (8.7%)
Phototherapy	13 (8.07%)
NICU admission	10 (6.21%)
Maternal morbidity	8 (4.97%)
Skin-to-skin contact (N,%)	33 (20.5%)
Breast pump during hospital stay (N,%)	151 (93.79%)
Breast pump frequency (N,%)	
Every 3 hours	76 (50.33%)
1–2 times/day	45 (29.8%)
3–4 times/day	25 (16.56%)
Other	5 (3.31%)
Breast pump at home (N,%)	99 (61.49%)

**Table 4 nutrients-14-00140-t004:** Nutrition characteristics of the studied babies.

Characteristic	N (%)
First feeding of the infant	
Breastfed	105 (65.2%)
Formula	35 (21.74%)
IV fluids	21 (13.04%)
Reasons for formula administration	
Weight loss	58 (44.27%)
Maternal desire	21 (16.03%)
NICU admittance	15 (11.45%)
Phototherapy	13 (9.92%)
Twin pregnancy	12 (9.16%)
Mother’s morbidity	12 (9.16%)
Breastfeeding duration	
1 week	161 (100%)
1st month	159 (98.8%)
4th month	140 (87%)
6th month	91 (56.5%)
12th month	66 (41%)
>12th month	31 (19.3%)

**Table 5 nutrients-14-00140-t005:** Weight gain at 1st, 4th, 6th month of age in relation to gender, birth weight (BW) group and gestational age (GA) group.

Weight Gain in Grams	Males (N = 95)	Females (N = 66)	
1st month	1048 ± 759	1017 ± 612	
4th month	3748 ± 850	3211 ± 710	
6th month	4975 ± 899	4369 ± 801	
Weight gain (percentage)			
1st month	34% ± 26%	36% ± 24%	
4th month	125% ± 44%	114% ± 36%	
6th month	165% ± 52%	157% ± 47%	
**Weight Gain in Grams**	**BW < 2500 gr (N = 30)**	**BW 2500–3500 gr (N = 90)**	**BW ≥ 3500 gr (N = 41)**
1st month	1159 ± 803	955 ± 650	1118 ± 731
4th month	3872 ± 877	3426 ± 854	3475 ± 704
6th month	5073 ± 979	4668 ± 921	4565 ± 768
Weight gain (percentage)			
1st month	51% ± 36%	32% ± 21%	30% ± 20%
4th month	169% ± 39%	116% ± 33%	92% ± 19%
6th month	223% ± 43%	158% ± 40%	121% ± 21%
**Weight Gain in Grams**	**GA < 37 weeks (N = 39)**	**GA ≥ 37 weeks (N = 122)**	
1st month	1065 ± 742	1026 ± 691	
4th month	3938 ± 830	3374 ± 788	
6th month	5303 ± 840	4526 ± 846	
Weight gain (percentage)			
1st month	44% ± 32%	32% ± 22%	
4th month	161% ± 39%	105% ± 30%	
6th month	220% ± 41%	141% ± 35%	

## Data Availability

Data are available from the corresponding autor upon a reasonale request.

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
