# Peer review of "Mothers in Need of Lactation Support May Benefit from Early Postnatal Galactagogue Administration: Experience from a Single Center"

_nutrients, 2021, doi:10.3390/nu14010140_

Round 1

Reviewer 1 Report

Title: MOTHERS IN NEED OF LACTATION SUPPORT MAY BENEFIT FROM EARLY POSTNATAL GALACTAGOGUE ADMINISTRATION: EXPERIENCE FROM A SINGLE CENTRE

The study focused on the very important aspects of newborn nutrition. The paper is original, however, there are several concerns that limit the validity of this work. Please address the following issues:

  • The aim of the study was (lines 12-13) to investigate the efficacy of Silitidil to increase milk 12 supply and duration of breastfeeding of mothers in need of lactation support and (lines 72-73) to assess whether the administration of Silitidil (Silymarin from thistle with incorporated phospholipids) was able to initiate/establish, maintain and increase milk supply and influence the duration of breastfeeding of mothers in need of lactation support. However, in my opinion on the basis of the described methodology and presented results, the effectiveness of this galactagogue cannot be assessed.
  • The authors state that mothers with insufficient milk supply were included in the study (lines 142-143). Please explain how that was evaluated.
  • Please describe whether and what kind of support systems were used to improve breastfeeding effectiveness in the hospital.
  • Mothers included in the study were advised to take a single dose of 5 g of product for at least 14 days. The authors do not provide detailed information about the actual dose taken, time of intake, or length of use. In addition, it is worth clarifying whether the preparate was offered for free or whether mothers purchased the formula.
  • The only parameter used for the assessment of the effectiveness of the preparation was the child's weight gain. However, this information comes from a telephone interview and may bear a large dose of uncertainty.
  • The results concerning weight gain of newborns were presented as median, Q1 and Q3 (table 3). In my opinion, the results should be presented according to the groups distinguished in Table 2 (birth weight). Please explain whether the infant weight data deviated from a normal distribution. If not, it would be useful to provide the mean and standard deviation/standard error.
  • Finally, objective measurement of milk volume and statistical analysis, including confounding factors are essential to conclude on the effectiveness of galactagogue use (lines 22-25, 196-198).

In conclusion, the article requires major revisions concerning methodology, interpretation of the results and language quality.

Author Response

  1. The authors state that mothers with insufficient milk supply were included in the study (lines 142-143). Please explain how that was evaluated.

  1. Please describe whether and what kind of support systems were used to improve breastfeeding effectiveness in the hospital.

  1. Mothers included in the study were advised to take a single dose of 5 g of product for at least 14 days. The authors do not provide detailed information about the actual dose taken, time of intake, or length of use. In addition, it is worth clarifying whether the preparate was offered for free or whether mothers purchased the formula.

  1. The only parameter used for the assessment of the effectiveness of the preparation was the child's weight gain. However, this information comes from a telephone interview and may bear a large dose of uncertainty.

  1. The results concerning weight gain of newborns were presented as median, Q1 and Q3 (table 3). In my opinion, the results should be presented according to the groups distinguished in Table 2 (birth weight). Please explain whether the infant weight data deviated from a normal distribution. If not, it would be useful to provide the mean and standard deviation/standard error.

  1. Finally, objective measurement of milk volume and statistical analysis, including confounding factors are essential to conclude on the effectiveness of galactagogue use (lines 22-25, 196-198).

Dear Reviewer,

Thank you for your thoughtful comments and considerations regarding our manuscript. We would like to kindly answer the issues you addressed.

  1. To begin with, our sample included mothers that had a higher risk of not achieving exclusive breastfeeding according to recent studies such as twin pregnancies, prematurity or severe illness of the mother. All those are a risk factors for inhibiting the initiation and continuation of breastfeeding. In case of excessive weight loss of the infant (>10% of BW) and phototherapy of the neonate, mothers were encouraged to pump during their hospital stay in order to evaluate milk supply. If insufficient milk supply was noted, galactagogues were given.

  1. Furthermore, because our hospital is baby-friendly it provides a support system regarding breastfeeding. Healthcare professionals receive continuous courses and seminars regarding breastfeeding. They assist with breastfeeding initiation (good position of the infant and good latch). We also implement rooming-in, have no advertisement or promotion of infant formulas and provide frequent breast pumping if indicated to enhance milk supply. Neonates are put skin to skin with the mother right after birth and breastfeed during the first hour of life (golden hour).

  1. The 5 gr of galactagogue was offered for free to the mothers that were included in the study during their hospital stay. After that period of time we prescribed the galactagogues and they were purchased by the mothers. The prescripted galactagogues were taken once a time daily for 14 days.

  1. According to national guidelines by the institute of child health, infants are vaccinated and clinically evaluated at specific age milestones (clinical assessment, weight, height, head circumference etc.) by a certified health care professional (pediatrician). Those data are included at each child’s personal health record. As far as the weight of the infant is concerned, the data was extracted by the mothers from the personal health record of the infant at those specific age milestones. In this way, although conducted through telephone interview, the information provided are certain.

5. Thank you for pointing, test for normality was performed via the Kolmogorov- Smirnov test, birth weight and maternal age were found to be derived from a normal distribution, thus the descriptive values for the arithmetic data were expressed as median mean±SD (Standard Deviation). We have changed the description in the statistics section to reflect this. Moreover, table 5 is augmented to include additional groupings as presented in table 2, and instead of median and Q1-Q3 range is now presented the mean±SD.

  1. Regarding the objective measurement of milk volume, pumping was used only during the hospital stay in order to enhance milk supply. Because the duration of our study was one year, the compliance of the mothers regarding daily pumping would be low and the difference at the time of pumping during the day and the distance between pumping and breastfeeding could act as an additional confounding factor. We chose to evaluate the sufficient milk supply by measuring the weight of the exclusively breastfed infant at the first six months of life and the duration of breastfeeding after that period. We recognize that our study has confounding factors and that larger studies with a bigger sample are needed in order to confirm the efficacy of the galactagogues.

Reviewer 2 Report

Thanks for working on this very relevant study. Please see my comments for your consideration in the attachment. 

Thanks. 

Author Response

PAGE

LINE

COMMENT

INTRODUCTION

2

64-65

Sentence rephrased

66

Words deleted

73

Word changed

MATERIALS AND METHODS

2

94

Change is made

RESULTS

3

98-101

Change is made

108

Change is made

5

131

Change is made

DISCUSSION

5

146

Change is made

161-162

Rephrased

Word added

167

Word changed

172-173

Sentence rephrased. Our goal was to describe that in our study, despite the high chances of breastfeeding discontinuation, we achieved a rate of 100% breastfeeding during the first week of life, higher than the national rate of 90%

6

177

Word changed

187

Word changed

CONLCUSION

6

198

Word changed

Thank you to the Reviewer for the thoughtful and useful comments.

Round 2

Reviewer 2 Report

Thanks for addressing the comments suggested sufficiently. 

Please check further comments below for your consideration:

Abstract: £37 to <37

Introduction: (second paragraph). Please use "..." in the phrase: low breast milk supply. Instead of <<...>>

Materials and Methods and in the Results (Table 5):

Use g and not gr as an SI unit for grams.